# Speckles and paraspeckles coordinate to regulate HSV-1 genes transcription

Kun Li[1] & Ziqiang Wang [1,2] ✉

Numbers of nuclear speckles and paraspeckles components have been demonstrated to regulate herpes simplex virus 1 (HSV-1) replication. However, how HSV-1 infection affects the two nuclear bodies, and whether this influence facilitates the expression of viral genes, remains elusive. In the current study, we found that HSV-1 infection leads to a redistribution of speckles and paraspeckles components. Serine/arginine-rich splicing factor 2 (SRSF2), the core component of speckles, was associated with multiple paraspeckles components, including *nuclear paraspeckles assembly transcript 1* (*NEAT1*), PSPC1, and P54nrb, in HSV-1 infected cells. This association coordinates the transcription of viral genes by binding to the promoters of these genes. By association with the enhancer of zeste homolog 2 (EZH2) and P300/CBP complex, *NEAT1* and SRSF2 influenced the histone modifications located near viral genes. This study elucidates the interplay between speckles and paraspeckles following HSV-1 infection and provides insight into the mechanisms by which HSV-1 utilizes host cellular nuclear bodies to facilitate its life cycle.

[1] Department of Nuclear Medicine, The First Affiliated Hospital of Shandong First Medical University & Shandong Provincial Qianfoshan Hospital, Jinan 250014, China. [2] Biomedical Sciences College & Shandong Medicinal Biotechnology Centre, Shandong First Medical University & Shandong Academy of Medical Sciences, Jinan 250062, China. ✉email: yky2009@163.com

In mammalian cells, numerous granular structures exist in the nucleus, including the speckles, paraspeckles, nucleoli, cajal bodies, PML nuclear bodies, and polycomb bodies, among others. These nuclear bodies are involved in multiple biological processes, including genes transcription, RNA processing, ribosome biogenesis, chromatin modification, and protein degradation[1,2]. Nuclear speckles are components of the spliceosome and participate in pre-mRNA splicing and the output of mRNAs. Serine/arginine-rich splicing factor 2 (SRSF2) contributes to the integrity of the nuclear speckles and functions to maintain genome stability, regulate pre-mRNA splicing, mRNAs output, and translation[3–7]. Paraspeckles are subnuclear structures that are located adjacent to speckles[8], and are built on specific long non-coding RNAs (lncRNAs), *nuclear paraspeckles assembly transcript 1* (*NEAT1*). *NEAT1* functions as a scaffolding RNA to binding multiple RNA-binding proteins, including P54nrb and PSPC1, to form paraspeckles. The primary function of paraspeckles is to mediate the nuclear retention of some A-to-I hyper-edited mRNAs, gene transcription, RNA splicing, and RNA stability[9–11]. In the case of speckles and paraspeckles, they are adjacent in spatial location and collaborate in their function of precursor mRNA splicing.

Herpes simplex virus 1 (HSV-1) infection is prevalent, with approximately 80−90% of the population having been infected with HSV-1 in their lifetime[12]. HSV-1 infection has been reported to be closely related to numerous diseases, including herpes labialis, genital herpes[13], Alzheimer's disease[14], encephalitis[15], and others. HSV-1 is a DNA virus that possesses a double-stranded DNA genome encoding over 84 viral proteins[16]. A growing body of evidence has suggested that HSV-1 exploits a series of host cell factors in order to facilitate its life cycle[17–20]. Our previous studies demonstrated that components of the speckles, SRSF2, and components of the paraspeckles, *NEAT1*, P54nrb, and PSPC1, also function to modulate HSV-1 replication and viral genes expression by binding these genes in order to alter their transcriptional activity[21,22]. Evidence has been described that HSV-1 infection causes a redistribution of speckles components[23] to inhibit host cell splicing[24–27]. However, little is known regarding the effects of this redistribution on the spatial location between the speckles and paraspeckles.

In the current study, we revealed the interplay between the speckles and paraspeckles in HSV-1 infection. We found that HSV-1 infection acts as the external stimulus, resulting in the redistribution of the speckles and paraspeckles components. In HSV-1 infected cells, SRSF2, the component of the speckles, is shown to co-localize with multiple components of the paraspeckles, including *NEAT1*, P54nrb, and PSPC1. *NEAT1* acts as the scaffolding RNA and influences the binding ability of SRSF2 to viral genes. In addition, *NEAT1* and SRSF2 collaborate to regulate the histone modification of nearby viral genes through the association with enhancer of zeste homolog 2 (EZH2) and the P300/CBP complex.

## Results

**HSV-1 infection redistributed speckles and paraspeckles.** Given that the speckles and paraspeckles are two adjacent subcellular organelles in a special location and the component of speckles, and SRSF2, the components of paraspeckles, *NEAT1*, PSPC1, and P54nrb, function as transcriptional activators for HSV-1 genes expression reported by our previous studies[21,22], we hypothesized that the components of the speckles and paraspeckles work together to modulate viral genes transcription under HSV-1 infection conditions. In order to determine whether HSV-1 infection resulted in a reorganization of these components, we first confirmed the infection efficiency of HSV-1 in HeLa cells

(Supplementary Fig. 1a) and C-33A cells (Supplementary Fig 1b), and then performed an RNA fluorescence in situ hybridization (FISH) and immunofluorescence assay to examine the interaction between *NEAT1*, PSPC1, P54nrb, and SRSF2 in HSV-1 or Mock infected HeLa and C-33A cells. The results demonstrated that HSV-1 infection promotes the co-localization of SRSF2 with the paraspeckles components *NEAT1*, PSPC1, and P54nrb (Fig. 1a and Supplementary Fig. 2a). In addition, we quantified the Pearson's coefficient and Overlap coefficient between channels to measure the degree of the colocalization using the plug-in, JACoP in ImageJ (Fig. 1b and Supplementary Fig. 2b). In order to determine the binding site(s), we performed an RNA immuno-precipitation (RIP) assay. The HeLa cells were infected with HSV-1 or Mock for 4 h. Then, the cell lysates were harvested and subjected to an immunoprecipitation assay using the SRSF2 antibodies. This was followed by qRT-PCR using primers that recognized *NEAT1* (Fig. 1c). These results demonstrated that HSV-1 infection significantly increased the fold enrichment of the *NEAT1*-N2 and *NEAT1*-N3 fragment pulled-down by SRSF2 antibodies, suggesting that *NEAT1*-N2 and *NEAT1*-N3 fragments have SRSF2-binding motifs (Fig. 1d). To confirm this association, we prepared probes by transcribing fragments of *NEAT1*-N2 in vitro. The probes were labeled with digoxigenin (DIG) at the 3′ end of the RNA and used to perform RNA FISH and immuno-fluorescence experiments in HSV-1 or Mock infected HeLa cells. An unrelated fragment *NEAT1*-N4 was used as a negative control. These results showed that HSV-1 infection promoted the colocalization between SRSF2 and *NEAT1*-N2 fragments (Fig. 1e). In addition, we quantified the Pearson's coefficient and Overlap coefficient between channels to measure the degree of the colocalization using the plug-in, JACoP in ImageJ (Fig. 1f). We also carried out an immunoprecipitation assay to confirm the association of SRSF2 with PSPC1 and P54nrb. HeLa cells were infected with HSV-1 or Mock for 4 h. The cell lysates were harvested and subjected to an immunoprecipitation assay using the SRSF2 antibodies, followed by western blotting using antibodies against PSPC1, P54nrb, and SRSF2. PSPC1 and P54nrb were slightly pulled down in the Mock infected cell lysates. HSV-1 infection was found to result in a significant increment in the association of SRSF2 with PSPC1 and P54nrb (Fig. 1g). These data demonstrated that HSV-1 infection led to a reorganization of speckles and paraspeckles.

**NEAT1 influences the association of SRSF2 with viral genes.** Our previous study found that SRSF2 and *NEAT1* positively regulated expression levels of HSV-1 *infected cell polypeptide 0* (*ICP0*), *thymidine kinase* (*TK*) genes[21,22]. Thus, we next investigated the roles of *NEAT1* and SRSF2 in the transcription of these two viral genes. We constructed a luciferase reporter containing the promoter fragment of *ICP0*, *TK*, *infected cell polypeptide 4* (*ICP4*), or *infected cell polypeptide 22* (*ICP22*) genes. *ICP4* gene and *ICP22* gene were used as negative controls. The luciferase assay demonstrated knockdown of *NEAT1* or SRSF2 with *NEAT1*-targeting siRNAs (Fig. 2a and Supplementary Fig. 3a) and SRSF2-targeting siRNAs (Fig. 2b and Supplementary Fig. 3b) both inhibited the transcriptional activities of the *ICP0* gene promoter and the *TK* promoter, with no significant effect on *ICP4* gene promoter and *ICP22* gene promoter in HSV-1-infected HeLa cells (Fig. 2c) and C-33A cells (Supplementary Fig. 3c). Because SRSF2 and *NEAT1* were reported to bind to these viral genes and the depletion of *NEAT1* was found to reduce the numbers of viral genes bound to P54nrb and PSPC1[21,22], we then investigated whether *NEAT1* influenced the ability of SRSF2 to bind to viral genes. To this end, we carried out a chromatin immunoprecipitation (ChIP) assay using an anti-SRSF2

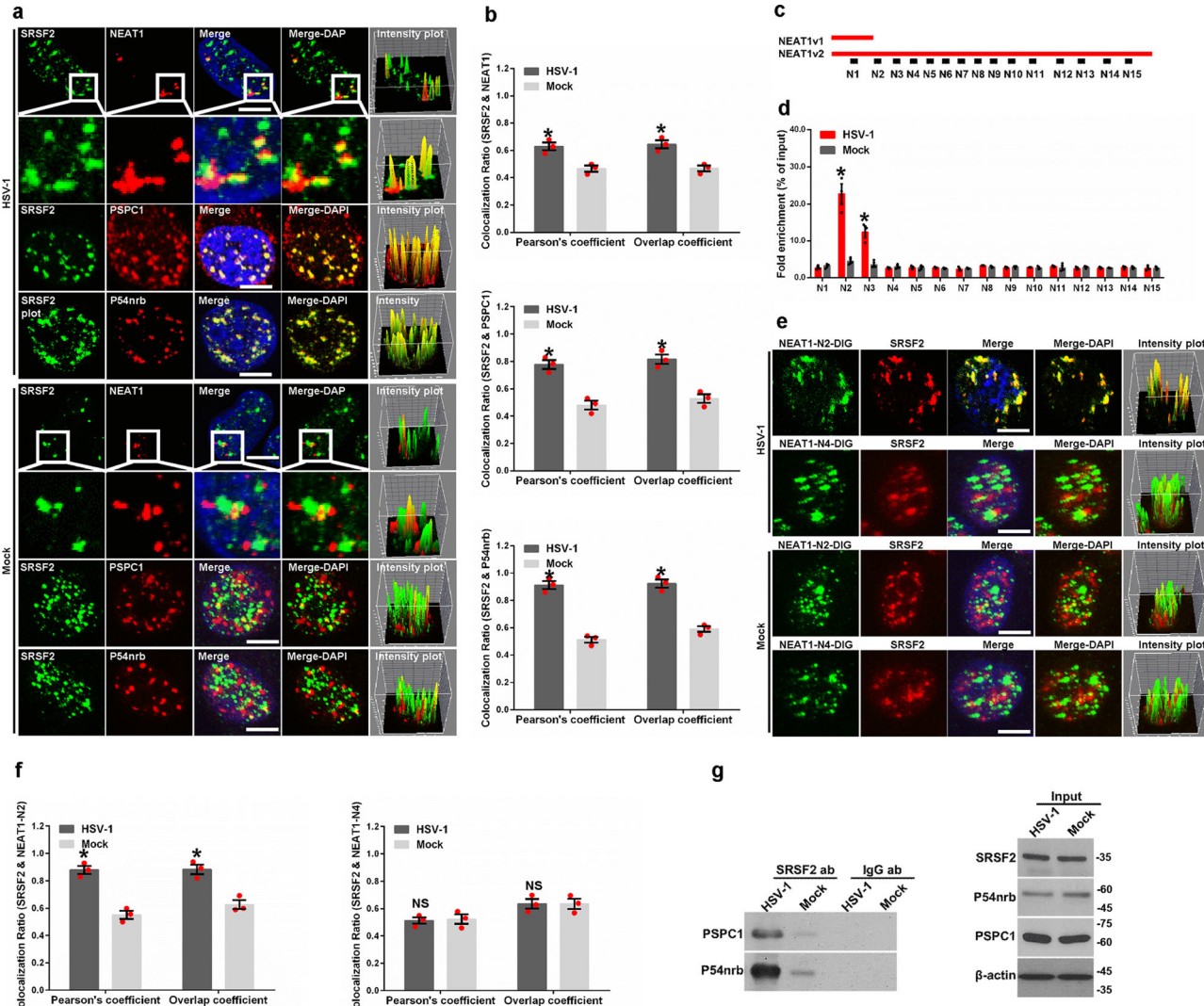

**Fig. 1 HSV-1 infection redistributes speckles and paraspeckles. a** HeLa cells infected with HSV-1 or Mock for 4 h were incubated with anti-SRSF2 antibodies (green) and then incubated with *NEAT1* probe (red), anti-PSPC1 antibodies (red), or anti-P54nrb antibodies (red). The images were captured with a confocal microscope. The intensity plots for the red and green channels were analyzed using ImageJ. DAPI (blue) was used to stain the nuclei. Scale bar, 10 μm. **b** The Pearson's coefficient and overlap coefficient for each merge channel in (**a**) were quantified using the JACoP in ImageJ. The data are presented as the mean ± standard deviation (SD) from three independent experiments (*$p < 0.01$, Student's *t*-test). **c** Schematic diagram showing the primer-amplified regions (black box) in the *NEAT1* sequence. **d** HeLa cells infected with HSV-1 or Mock for 4 h were harvested and subjected to a RIP assay. QRT-PCR was performed to detect the retrieval of *NEAT1* by anti-SRSF2 antibodies over the input level. The data are presented as the mean ± SD from three independent experiments (*$p < 0.01$, Student's *t*-test). **e** HeLa cells infected with HSV-1 or Mock for 4 h were fixed, incubated with DIG-labeled *NEAT1*-N2 fragment (green), *NEAT1*-N4 fragment (green), and SRSF2 (red), and subjected to confocal microscopy analysis. The intensity plots for the red and green channels were analyzed using Image J. DAPI (blue) was used to stain the nuclei. Scale bar, 10 μm. **f** The Pearson's coefficient and overlap coefficient of each merge channel in (**e**) were quantified for each merge channel using the JACoP in ImageJ. The data are presented as the mean ± SD from three independent experiments (*$p < 0.01$, Student's *t*-test, NS no significance). **g** HeLa cells were infected with HSV-1 or Mock for 4 h. The cell lysates were then harvested and subjected to an immunoprecipitation assay with anti-SRSF2 antibodies or anti-IgG antibodies. The retrieval of PSPC1 and P54nrb by endogenous SRSF2 or IgG and the input levels of SRSF2, PSPC1, and P54nrb were measured by western blotting.

antibodies in HSV-1-infected cells transfected with *NEAT1*-targeting siRNAs or negative control siRNAs. As expected, the enrichment of the SRSF2-associated *ICP0* gene and *TK* gene promoters was significantly reduced following the depletion of *NEAT1* in HSV-1-infected HeLa cells (Fig. 2d) and C-33A cells (Supplementary Fig. 3d), respectively. Next, we used DNA-FISH and immunofluorescence microscopy assays to validate the co-localization between SRSF2 and the HSV-1 genes. In order to visualize the HSV-1 genome, we first purified and sonicated HSV-1 genomic DNA. We then labeled the viral DNA with biotinylated cytidine bisphosphate at its 3′ end. These results confirmed

that SRSF2 was associated with the viral DNA, wherein the knockdown of *NEAT1* disrupted this co-localization (Fig. 2e). In addition, we generated pixel intensity plots for each merged channel (Fig. 2e, right panels) and quantified the Pearson's coefficient and Overlap coefficient between channels to measure the degree of the colocalization using the plug-in, JACoP in ImageJ (Fig. 2f). To confirm that this disruption was related to *NEAT1*, we performed an RNA-FISH assay with *NEAT1* staining to examine the transfection efficiency of *NEAT1*-targeting siRNAs in HSV-1-infected HeLa cells transfected with *NEAT1*-targeting siRNAs or negative control siRNAs. The results showed that cells

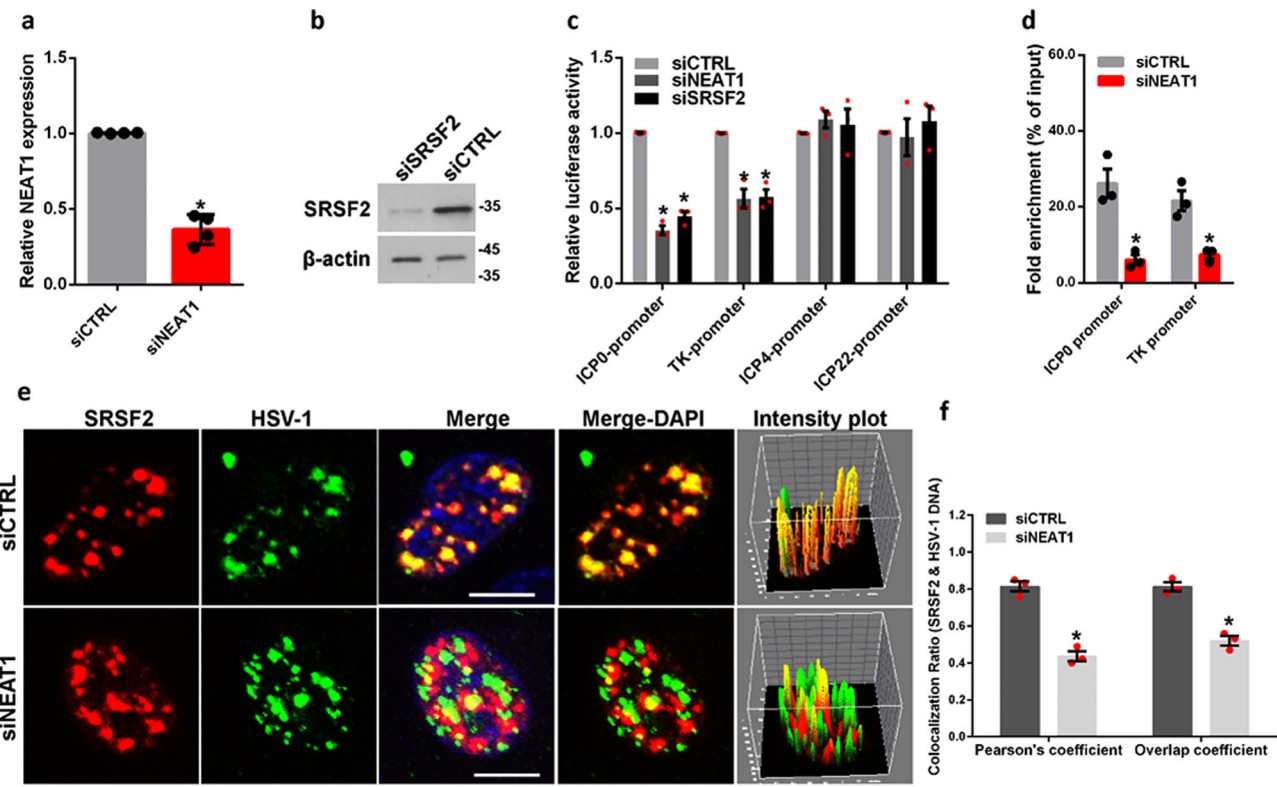

**Fig. 2 NEAT1 influences the binding ability of SRSF2 to viral genes. a** HeLa cells transfected with NEAT1-targeting siRNAs or negative control siRNAs were infected with HSV-1 for 4 h. The relative NEAT1 levels compared with β-actin mRNA were analyzed with real-time PCR. The data are presented as the mean ± SD from four independent experiments (*$p < 0.01$, Student's t-test). **b** HeLa cells transfected with the SRSF2-targeting siRNAs or negative control siRNAs were infected with HSV-1 for 4 h. The SRSF2 expression levels were measured using western blotting. **c** After co-transfection with NEAT1-targeting siRNAs, SRSF2-targeting siRNAs, or negative control and the pGL3 enhancer plasmid containing the ICP0 gene promoter, ICP4 gene promoter, ICP22 gene promoter, or a pRL-TK reporter, the HeLa cells were infected with HSV-1 for 4 h. The relative transcriptional activities were analyzed by luciferase assay. The data are presented as the mean ± SD from three independent experiments (*$p < 0.01$, Student's t-test). **d** HeLa cells transfected with NEAT1-targeting siRNAs or negative control siRNAs were infected with HSV-1 for 4 h. ChIP assays were performed with anti-SRSF2 antibodies, and the fold enrichment of the ICP0 and TK gene promoters by SRSF2 relative to the input level was examined using real-time PCR. The data are presented as the mean ± SD from three independent experiments (*$p < 0.01$, Student's t-test). **e** HeLa cells transfected with NEAT1-targeting siRNAs or negative control siRNAs were infected with HSV-1 for 4 h. The cells were then incubated with biotin-labeled HSV-1 genomic DNA probe (green) and anti-SRSF2 antibodies (red). The images were captured using a confocal microscope. The intensity plots for the red and green channels were analyzed using ImageJ. DAPI (blue) was used to stain the nuclei. Scale bar, 10 μm. **f** The Pearson's coefficient and overlap coefficient for each merge channel in (**e**) were quantified using the JACoP in ImageJ. The data are presented as the mean ± SD from three independent experiments (*$p < 0.01$, Student's t-test).

transfected with NEAT1-targeting siRNAs resulted in a significant decrease in the number of NEAT1 puncta per cell (Supplementary Fig. 4). Taken together, these results demonstrate that NEAT1, the scaffolding RNA of the paraspeckles, influences the association of SRSF2 with viral genes, suggesting an association between the speckles and paraspeckles in the transcription of viral genes.

**NEAT1 and SRSF2 influence histone modifications nearby viral genes.** Upon entry into the host cell nucleus, HSV-1 genomic DNA will package host cellular histones and epigenetically control the expression of viral genes[28–32]. In an effort to further understand the roles of the speckles and paraspeckles in viral gene transcription, we examined the histone modification status of the HSV-1 ICP0 and TK gene promoters in HSV-1-infected HeLa cells transfected with NEAT1-targeting siRNAs, SRSF2-targeting siRNAs, or negative control siRNAs. We performed ChIP experiments using antibodies against with histone H3, tri-methylated histone H3 at lysine 4 (H3K4Me3), tri-methylated histone H3 at lysine 27 (H3K27Me3), and acetylated histone H3 at lysine 27 (H3K27Ac). In these histone modifications, H3K4Me3 and H3K27Ac in transcription start sites (TSS) represent markers for actively transcribed genes, while

H3K27Me3 in TSS is associated with gene repression[33]. To identify these modified histone H3 binding sites within the sequences of the HSV-1 ICP0 and TK gene promoters, we designed a set of primer pairs that recognize the TSS regions of these genes (Supplementary Table 1). The ChIP assay demonstrated that the knockdown of NEAT1 or SRSF2 both resulted in a decreased enrichment of all three histone modifications at the promoters of these genes (Fig. 3a, b). In addition, we performed a DNA pull down assay to verify this association. The HeLa cells transfected with NEAT1-targeting siRNAs, SRSF2-targeting siRNAs, or negative control siRNAs were infected with HSV-1 for 4 h. The cell lysates were harvested and incubated with a PCR amplified biotin-labeled HSV-1 ICP0 gene promoter DNA probe. The DNA−protein complexes were pulled down using streptavidin beads. The bound proteins were resolved by SDS-PAGE and subsequently analyzed by western blotting. As expected, the recovery of H3K4Me3, H3K27Me3, and H3K27Ac was potently decreased from cells lysates with the knockdown of NEAT1 (Fig. 3c, d) or SRSF2 (Fig. 3e, f). In addition, we generated a scrambled sequence of ICP0 TSS-4 and performed a DNA pull down assay as a negative control (Supplementary Fig. 5). To examine the effect of NEAT1 and SRSF2 on the global levels of

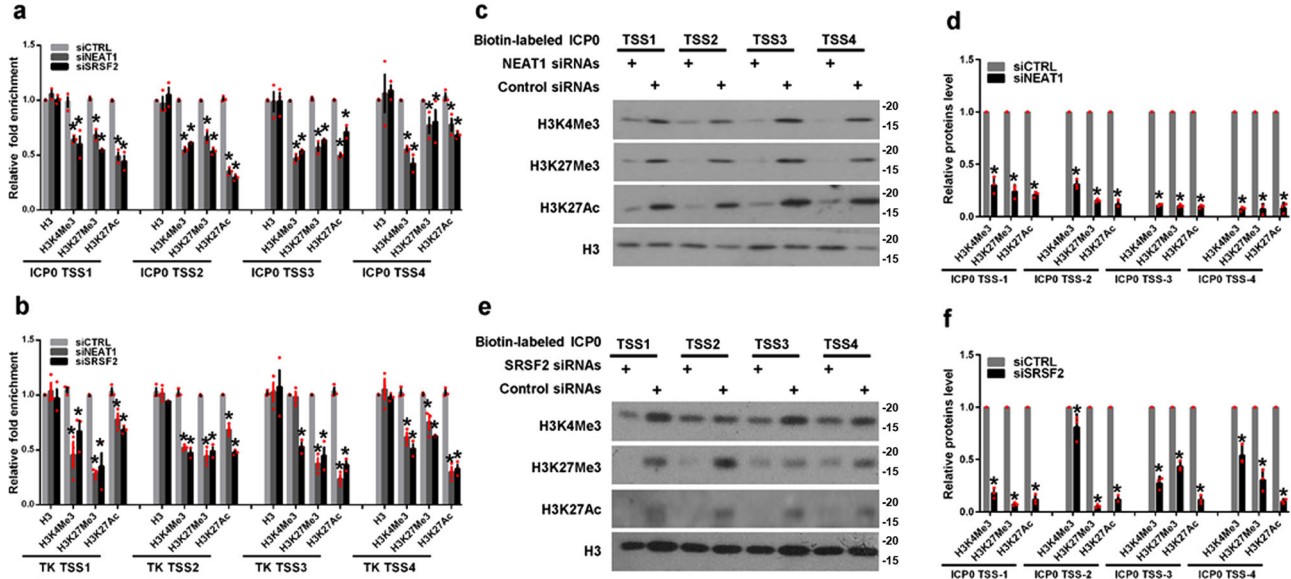

**Fig. 3 NEAT1 and SRSF2 influence histone modifications located near viral genes. a, b** HeLa cells transfected with *NEAT1*-targeting siRNAs, SRSF2-targeting siRNAs, or negative control siRNAs were infected with HSV-1 for 4 h. ChIP assays were performed with anti-histone H3 antibodies, anti-H3K4Me3 antibodies, H3K27Me3 antibodies, or H3K27Ac antibodies, and the fold enrichment of the *ICP0* (**a**) and *TK* (**b**) gene promoters relative to the input level was measured using real-time PCR. The data are presented as the mean ± SD from three independent experiments (*$p < 0.01$, Student's t-test). HeLa cells transfected with *NEAT1*-targeting siRNAs (**c**), SRSF2-targeting siRNAs (**e**), or negative control siRNAs were infected with HSV-1 for 4 h. The cell lysates were then harvested and incubated with a PCR amplified biotin-labeled HSV-1 *ICP0* gene promoter DNA probe. The DNA−protein complexes were pulled down by streptavidin beads. The bound proteins were then resolved using SDS-PAGE and subsequently analyzed by western blotting with anti-histone H3 antibodies, anti-H3K4Me3 antibodies, H3K27Me3 antibodies, or H3K27Ac antibodies. Protein ratios of the indicated proteins/histone H3 in Fig. 3c (**d**) and Fig. 3e (**f**) were analyzed with ImageJ software, and statistical analysis was conducted with data from three independent experiments. The data are presented as the mean ± SD (*$p < 0.01$, Student's t-test).

H3K4Me3, H3K27Me3, and H3K27Ac, we performed western blotting and immunofluorescence assay. The results demonstrated that the knockdown of *NEAT1* or SRSF2 both decreased the expressional level of H3K4Me3, H3K27Me3, and H3K27Ac in HSV-1 or Mock infected HeLa cells (Fig. 4a–d). These data suggest that *NEAT1* and SRSF2 function as epigenetic regulators of HSV-1 gene expression by altering the histone modifications located near these genes.

***NEAT1* and SRSF2 are associated with EZH2.** Given that enhancer of zeste homolog 2 (EZH2), the component of polycomb repressive complex 2 (PCR2), is a methyltransferase for the methylation of histone H3 at lysine 27[34] and *NEAT1* was previously reported to bind to EZH2 to mediate the trimethylation of H3K27 in glioblastoma cells[35], we performed an RNA-FISH assay to determine the relationship between *NEAT1*, SRSF2, and EZH2. The result demonstrated that *NEAT1* and SRSF2 were found to both co-localize with EZH2 (Fig. 5a). In addition, each merged channel generated by ImageJ (Fig. 5a, right panels) and the Pearson's coefficient and Overlap coefficient between channels quantified by the plug-in, JACoP in ImageJ (Fig. 5b) suggested the co-localization between *NEAT1*, SRSF2, and EZH2. To confirm the association between *NEAT1* and EZH2, we carried out an RNA immunoprecipitation assay using antibodies against EZH2 or IgG, followed by qRT-PCR using primers designed to recognize different *NEAT1* fragments (Fig. 1c). As shown in Fig. 5c, the antibodies against to EZH2 pulled down the fragments 2, 3, 4, 8, 10, and 14 of *NEAT1*. In addition, an immunoprecipitation assay was performed to verify the relationship between SRSF2 and EZH2. The cell lysates were harvested from HSV-1-infected HeLa cells that were transiently transfected with SRSF2-targeting siRNAs or negative control siRNAs. These lysates were then subjected to an immunoprecipitation assay using antibodies against

SRSF2, followed by western blotting using antibodies against EZH2 and SRSF2. These results demonstrated that EZH2 were indeed specifically pulled down by SRSF2, and that the silencing of SRSF2 expression using specific SRSF2-targeting siRNAs resulted in a reduction in the levels of EZH2 pulled down by SRSF2 (Fig. 5d). These data suggest that *NEAT1* and SRSF2 are involved in the methylation of H3K27 through an association with EZH2.

***NEAT1* and SRSF2 are associated with P300/CBP complex.** Since studies have reported that the acyl-transferases P300/CBP complex is involved in histone acetylation and that it epigenetically regulates related gene transcription[36,37], an RNA-FISH assay was performed to determine the relationship between *NEAT1*, SRSF2, and P300/CBP with regards to spatial location. As shown in Fig. 6a, *NEAT1* and SRSF2 were found to both largely co-localize with P300 and CBP. In addition, we quantified the Pearson's coefficient and Overlap coefficient between channels to measure the degree of the colocalization using the plug-in, JACoP in ImageJ (Fig. 6b). To confirm the association between the *NEAT1* and P300/CBP complex, an RNA immunoprecipitation assay was carried out using antibodies against P300, CBP, or IgG followed by qRT-PCR using primers designed to recognize different *NEAT1* fragments (Fig. 1c). The results showed that antibodies against P300 pulled down fragments 2, 3, 8, 10, 11, 12, and 13 of *NEAT1*, and antibodies against CBP pulled down fragments 3, 8, 10, 11, 12, and 13 of *NEAT1* (Fig. 6c). In addition, an RNA-FISH assay with staining P300, CBP, and the 5′ segment of *NEAT1v2* (1-3756), which have no binding sites in the interaction of *NEAT1* with P300 and CBP as a negative control. The results showed that there is no co-localization of *NEAT1v2* (1-3756) with P300 and CBP (Supplementary Fig. 6). To verify the relationship between SRSF2 and the P300/CBP complex, we performed an

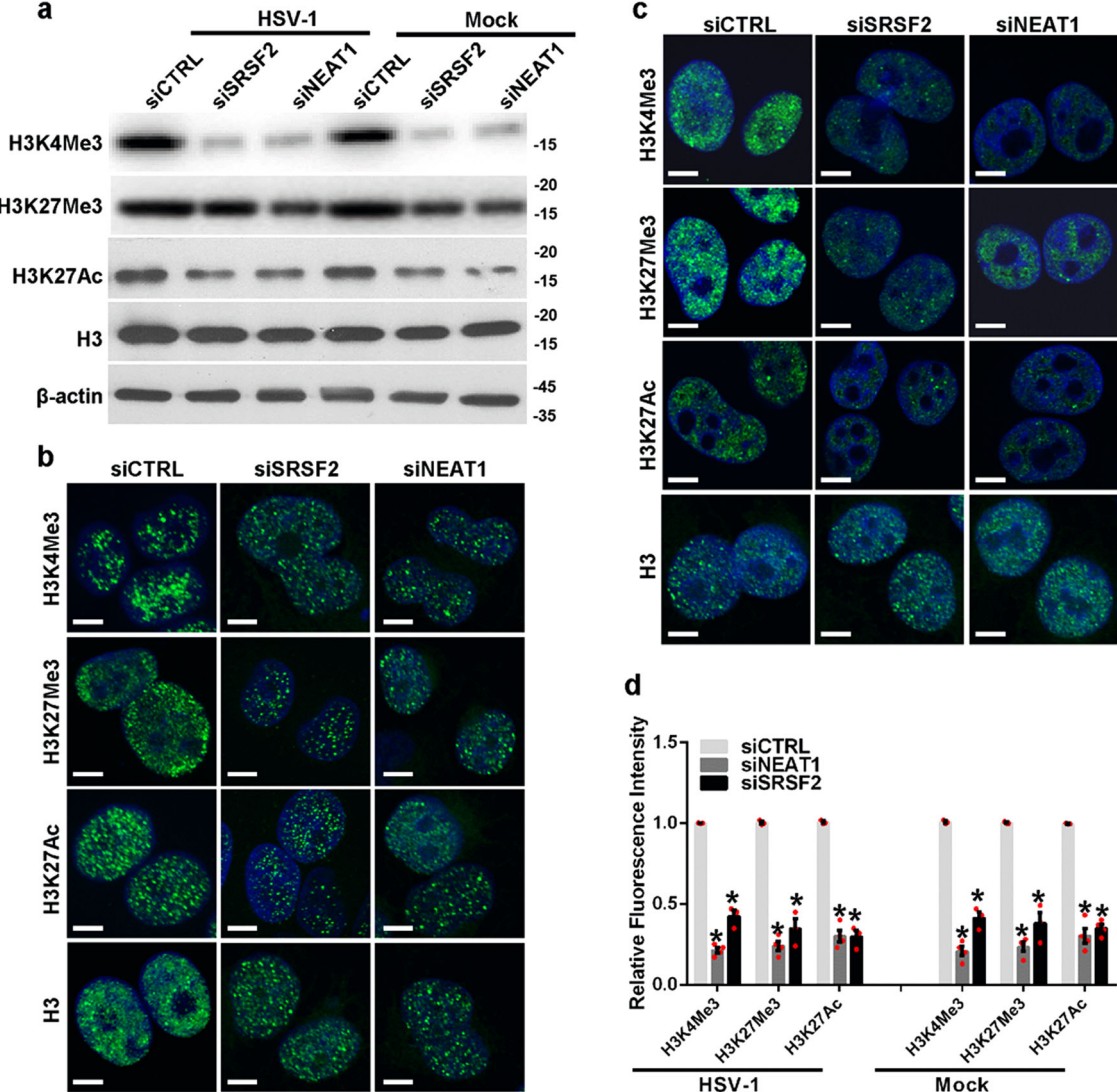

**Fig. 4 NEAT1 and SRSF2 regulates the global levels of histone modifications. a** HeLa cells transfected with *NEAT1*-targeting siRNAs, SRSF2-targeting siRNAs, or negative control siRNAs were infected with HSV-1 or Mock for 4 h. The H3K4Me3, H3K27Me3, H3K27Ac, and histone H3 expression levels were measured with western blotting. HeLa cells transfected with *NEAT1*-targeting siRNAs, SRSF2-targeting siRNAs, or negative control siRNAs were infected with HSV-1 (**b**) or Mock (**c**) for 4 h. The cells were then incubated with anti-H3K4Me3 antibodies, anti-H3K27Me3 antibodies, anti-H3K27Ac antibodies, or anti-Histone H3 antibodies. The images were captured with a confocal microscope. DAPI (blue) was used to stain the nuclei. Scale bars, 10 μm. **d** The relative fluorescence intensity in (**b**, **c**) was analyzed using ImageJ software. The data are presented as the mean ± SD from four independent experiments (*$P < 0.01$, Student's t-test).

immunoprecipitation assay using antibodies against SRSF2, followed by western blotting using antibodies against P300, CBP, and SRSF2. The results demonstrated that P300 and CBP were indeed specifically pulled down by SRSF2, and silencing SRSF2 expression resulted in a reduction in the levels of P300 and CBP pulled down by SRSF2 (Fig. 6d). These data suggest that *NEAT1* and SRSF2 are involved in the acetylation of H3K27 through the association with the P300/CBP complex.

## Discussion

In mammalian cells, nuclear bodies are present at a steady-state and have been shown to dynamically respond to various external stimulators. These stimulations often result in the redistribution of nuclear architecture[38–42]. Given that HSV-1 infection leads to a reorganization of the speckles[23], and that both the speckles and the parapspeckles that are adjacent to the speckles were reported by our previous studies to play a role in HSV-1 viral genes expression[21,22],

we hypothesized that HSV-1 infection could influence the spatial localization between the speckles and parapspeckles. As expected, we showed that SRSF2, the core component of the speckles, co-localized with the three primary components of the parapspeckles, *NEAT1*, PSPC1, and P54nrb, in HSV-1 infected cells.

We next investigated the effects of the redistribution of speckles and parapspeckles on viral genes expression. While both SRSF2, *NEAT1*, P54nrb, and PSPC1 were reported to regulate HSV-1 genes transcription by binding to these viral genes, and the depletion of *NEAT1* was shown to reduce the numbers of viral genes bound to P54nrb and PSPC1[21,22], we determined the effect of *NEAT1* on the binding ability of SRSF2 to viral genes. Our results demonstrated that depletion of *NEAT1* results in a significant reduction in the interaction of SRSF2 with viral genomic DNA. This suggests that *NEAT1* functions as a scaffolding RNA to tether SRSF2 to viral genes for these genes transcription.

Since histone modification plays a vital role in epigenetic control of gene expression by changing the configuration of

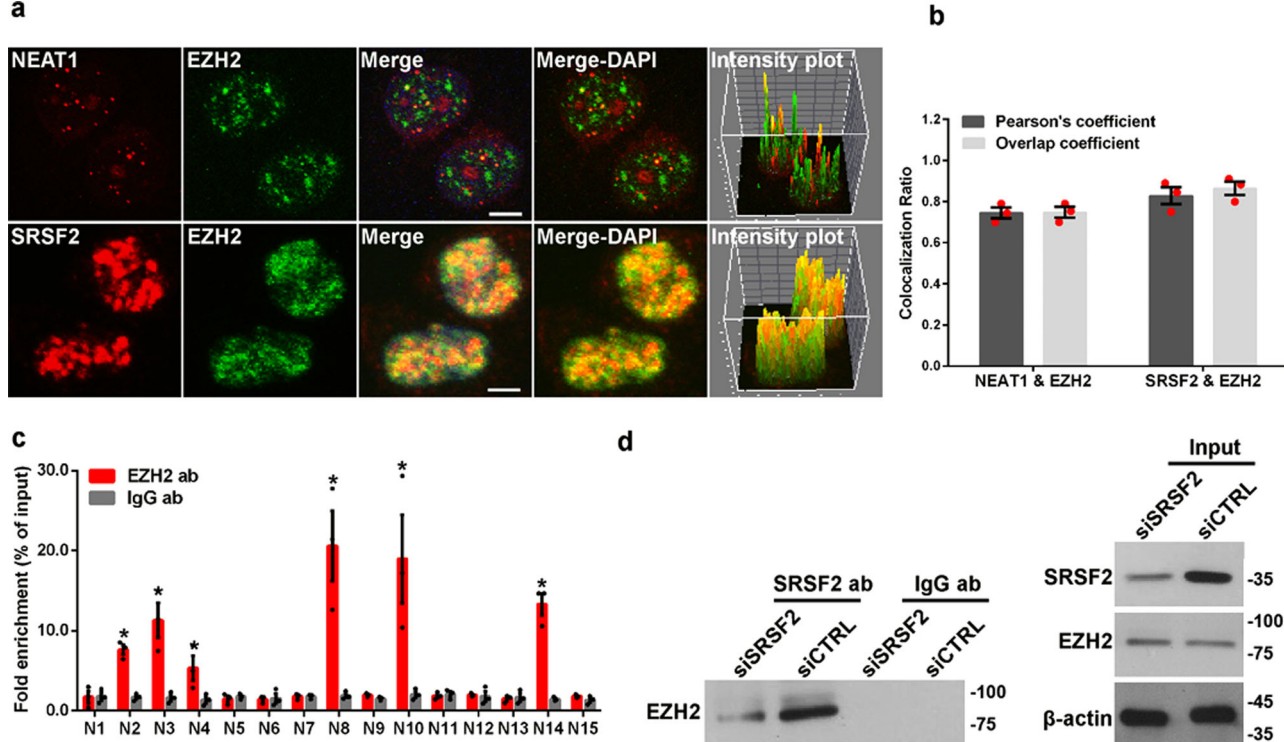

**Fig. 5 NEAT1 and SRSF2 are associated with EZH2. a** HeLa cells infected with HSV-1 for 4 h were incubated with anti-SRSF2 antibodies (red) or *NEAT1* probe (red), then incubated with anti-EZH2 antibodies (green). The images were captured using a confocal microscope. The intensity plots for the red and green channels were analyzed using ImageJ. DAPI (blue) was used to stain the nuclei. Scale bar, 10 μm. **b** The Pearson's coefficient and overlap coefficient for each merge channel in (**a**) were quantified using the JACoP in ImageJ. The data are presented as the mean ± SD from three independent experiments (*$p < 0.01$, Student's *t*-test). **c** HeLa cells infected with HSV-1 for 4 h were harvested and subjected to a RIP assay. QRT-PCR was performed to detect the retrieval of *NEAT1* by anti-EZH2 antibodies or anti-IgG antibodies over the input level. The data are presented as the mean ± SD from three independent experiments (*$p < 0.01$, Student's *t*-test). **d** HeLa cells *t*ransfected with SRSF2-targeting siRNAs or negative control siRNAs were infected with HSV-1 for 4 h. The cell lysates were then harvested and subjected to an immunoprecipitation assay with anti-SRSF2 antibodies or anti-IgG antibodies. The retrieval of EZH2 by endogenous SRSF2 or IgG and the input levels of SRSF2 and EZH2 were measured by western blotting.

chromatin and the recruitment of transcription factors to gene promoters[43–45], we investigated the effects of the interplay between the speckles and paraspeckles in HSV-1 infection on the methylation and acetylation status of the viral genome. We found that the core components of the speckles and paraspeckles, SRSF2 and *NEAT1*, worked together to regulate histone modifications nearby *ICP0* and *TK* genes' TSS, including H3K4Me3, H3K27Me3, and H3K27Ac. In these altered histone modifications, H3K4Me3 and H3K27Ac represent markers for actively transcribed genes, while H3K27Me3 is associated with gene repression. We think the role of *NEAT1* and SRSF2 in viral genes' transcriptional activity is resulted from the comprehensive effect of these altered histone modifications. To elucidate the mechanism by which SRSF2 and *NEAT1* modulate these histone modifications at viral genes promoters, we first confirmed the effect of SRSF2 and *NEAT1* on the global levels of these histone modifications. Given that the polycomb group protein EZH2 and the acyl-transferases P300/CBP complex are involved in the methylation and acetylation at H3K27, respectively[34,36,37], we then determined the relationship between *NEAT1*, SRSF2, EZH2, and P300/CBP in regards to spatial location and observed their association. This suggests that *NEAT1* and SRSF2 function as epigenetic regulators in related genes expression through the association with EZH2 and P300/CBP complex.

In the present study, we found that HSV-1 infection promoted the colocalization between the core component of the speckles SRSF2 and the primary components of the paraspeckles *NEAT1* and paraspeckle proteins (PSPCs). Then, through binding with

HSV-1 genes promoters and associating with acyltransferase P300/CBP and EZH2, SRSF2 and *NEAT1* coordinated to alter histone modifications nearby viral genes, finally increasing these genes transcription (Fig. 6e). Taken together, our current study has revealed the interplay between speckles and paraspeckles in HSV-1 infection. The epigenetic regulation of this interplay in viral genes expression provides a significant insight into how HSV-1 utilizes host cellular factors to facilitate the viral lifecycle.

## Methods

**Cell culture and HSV-1 infection**. HeLa cells (American Type Culture Collection, ATCC) and C-33A cells (ATCC) were grown in Dulbecco's modified Eagle's medium (Gibco/Invitrogen Ltd, 12800-017) containing 10% fetal bovine serum (PAA, A15-101) and 10 U/ml penicillin-streptomycin (Gibco/Invitrogen Ltd, 15140-122) in a humidified 5% $CO_2$ incubator at 37 °C. The HeLa and C-33A cells were infected with HSV-1 strain SM44 at an MOI of 1.

**Cells transfection, RNA isolation, reverse transcription, and qPCR**. All synthetic siRNAs were purchased from Shanghai GenePharma Co. Ltd. The sequences of the siRNAs used are listed in Supplementary Table 1. All siRNAs were transfected with lipofectamine™ 2000 (Invitrogen, 11668-019) according to the manufacturer's protocol. Total RNA was isolated using RNAiso Plus (Takara, D9108B) according to the manufacturer's protocol. Real-time qRT-PCR was performed using ReverTra Ace® qPCR RT Master Mix with gDNA remover (TOYOBO, FSQ-301) and SYBR Green PCR Master Mix (TOYOBO, QPK-201). All mRNA levels were measured and normalized to beta-actin. The primers for RT-PCR analysis are listed in Supplementary Table 1.

**Western blotting**. HeLa cells infected by HSV-1 were lysed in ice-cold whole-cell extract buffer B (50 mM TRIS-HCl, pH 8.0, 4 M urea, and 1% Triton X-100), supplemented with complete protease inhibitor mixture. Cell extracts resolved by

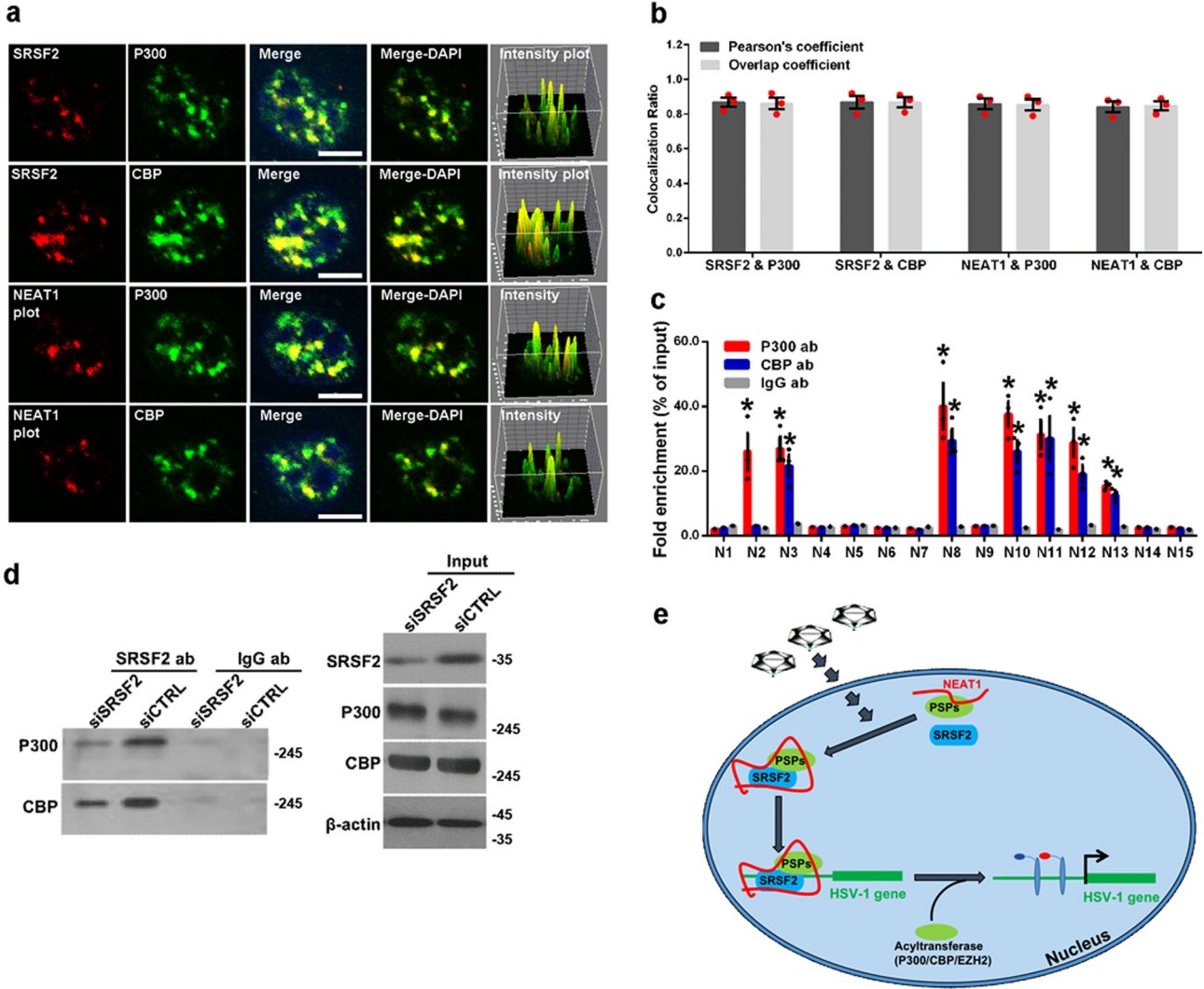

**Fig. 6 *NEAT1* and SRSF2 are associated with the P300/CBP complex. a** HeLa cells infected with HSV-1 for 4 h were incubated with anti-SRSF2 antibodies (red) or *NEAT1* probe (red), and then incubated with anti-P300 antibodies (green) or anti-CBP antibodies (green). The images were captured using a confocal microscope. The intensity plots for the red and green channels were analyzed using ImageJ. DAPI (blue) was used to stain the nuclei. Scale bar, 10 μm. **b** The Pearson's coefficient and overlap coefficient for each merge channel in (**a**) were quantified using the JACoP in ImageJ. The data are presented as the mean ± SD from three independent experiments (*$p < 0.01$, Student's *t*-test). **c** HeLa cells infected with HSV-1 for 4 h were harvested and subjected to a RIP assay. QRT-PCR was performed to detect the retrieval of *NEAT1* by anti-P300 antibodies, anti-CBP antibodies, or anti-IgG antibodies over the input level. The data are presented as the mean ± SD from three independent experiments (*$p < 0.01$, Student's *t*-test). **d** HeLa cells *t*ransfected with SRSF2-targeting siRNAs or negative control siRNAs were infected with HSV-1 for 4 h. The cell lysates were then harvested and subjected to an immunoprecipitation assay with anti-SRSF2 antibodies or anti-IgG antibodies. The retrieval of P300 and CBP by endogenous SRSF2 or IgG and the input levels of SRSF2, P300, and CBP were measured using western blotting. **e** Schematic model of the role of speckle and paraspeckle components in viral genes transcription.

SDS-PAGE were analyzed by western blotting. Protein bands were visualized using ECL Blotting Detection Reagents. Antibodies used for western blotting include anti-H3K4Me3 antibodies (Abcam, ab8580), anti-H3K27Me3 antibodies (Abcam, ab6002), anti-H3K27Ac antibodies (Abcam, ab4729), anti-histone H3 antibodies (Abcam, ab1791), anti-SRSF2 (Santa Cruz Biotechnology, sc-10252), and anti-beta-actin antibodies (Proteintech, 60008-1-Ig).

**RNA/DNA FISH**. To determine the interaction of *NEAT1* with SRSF2, EZH2, P300, or CBP in HSV-1 infection, RNA-FISH was performed as described previously[46]. Briefly, HSV-1 or Mock infected HeLa cells or C-33A cells were incubated with the *NEAT1* probe (Biosearch Technologies, SMF-2037-1) which consists of a set of Quasar® 570-labeled oligos against the middle segment (3800-11700 bp) of human *NEAT1v2* overnight at 37 °C and then incubated with anti-SRSF2 antibodies (Abcam, ab11826), anti-EZH2 antibodies (Abcam, ab228697), anti-P300 antibodies (Abcam, ab59240), or anti-CBP antibodies (Abcam, ab50702), for 1.5 h at room temperature. To confirm the transfection efficiency of *NEAT1*-targeting siRNAs, HSV-1 infected HeLa cells transfected with *NEAT1*-targeting siRNAs or negative control siRNAs were incubated with the *NEAT1* probe

(Biosearch Technologies, SMF-2037-1) overnight at 37 °C. To verify the interaction between *NEAT1* and SRSF2, the *NEAT1* N2 and N4 were transcribed in vitro by T7 RNA polymerase and labeled with digoxigenin (DIG) using the DIG Northern Starter Kit (Roche, 12039672910). HeLa cells infected with HSV-1 or Mock were incubated with the *NEAT1* N2 or N4 probe overnight at 37 °C and then incubated with anti-SRSF2 antibodies (Abcam, ab11826) for 1.5 h at room temperature. To determine the interaction of 5′ segment of *NEAT1v2* (1-3756) with P300 or CBP in HSV-1 infection, HeLa cells infected with HSV-1 were incubated with the *NEAT1v2* (1-3756) probe (Biosearch Technologies, SMF-2036-1) which consists of a set of Quasar® 570-labeled oligos against human *NEAT1v2* (1-3756 bp) overnight at 37 °C and then incubated with anti-P300 antibodies (Abcam, ab59240) or anti-CBP antibodies (Abcam, ab50702), for 1.5 h at room temperature. To study the role of *NEAT1* in the interaction between SRSF2 and HSV-1 genomic DNA, DNA-FISH was performed as described[22]. Briefly, HeLa cells were transfected with *NEAT1*-targeting siRNAs or negative control siRNAs for 36 h. Four hours after HSV-1 infection, the cells were incubated with biotin-labeled HSV-1 genomic DNA overnight at 42 °C and then incubated with anti-SRSF2 antibodies (Abcam, ab11826) for 1.5 h at room temperature. After the cells were washed and incubated

with the secondary antibodies, they were counterstained with DAPI and mounted for observation. Cell images were obtained with an Olympus FV1000 confocal microscope. The intensity plots for the red and green channels were analyzed using ImageJ. To estimate colocalization between channels, Pearson's coefficient and overlap coefficient were quantified in ImageJ using the plug-in, JACoP (Just Another Colocalization Plugin). Pearson's coefficient describes the correlation of the intensity distribution between channels and its values range between 1 (perfect positive correlation) and −1 (perfect negative correlation). Overlap coefficient represents the degree of colocalization and its values range between 0 (non-overlapping) and 1 (perfect colocalization)[47,48].

**Immunofluorescence assay.** Immunofluorescence assay was performed as described previously[49]. Briefly, HSV-1 infected HeLa cells or C-33A cells were incubated with indicated antibodies for 1.5 h at room temperature. After the cells were washed and incubated with the secondary antibodies, they were counterstained with DAPI and mounted for observation. Cell images were obtained with an Olympus FV1000 confocal microscope. The intensity plots for the red and green channels were analyzed using ImageJ. The fluorescence signal value was measured using ImageJ software and normalized to a single cell. To estimate colocalization between channels, Pearson's coefficient and overlap coefficient were quantified in ImageJ using the plug-in, JACoP.

**Luciferase assay.** For luciferase reporter assays, 100 ng luciferase reporter plasmids and 20 pmol siRNAs were co-transfected into $10^4$ HeLa cells in a 24-well culture plate. Cell lysates were harvested 24 h after transfection. Then luciferase activities were assayed with a Dual-Luciferase Reporter System following the manufacturer's instructions (Promega, E1960) and normalized to the total protein content of the cell lysate. The experiment was carried out in triplicate.

**ChIP assay.** ChIP assay was conducted as described previously in[50]. In brief, HSV-1 infected cells were fixed with 1% formaldehyde and sonicated to shear DNA. After centrifugation, the supernatants were incubated with anti-SRSF2 antibodies (Abcam, ab11826), anti-H3K4Me3 antibodies (Abcam, ab8580), anti-H3K27Me3 antibodies (Abcam, ab6002), anti-H3K27Ac antibodies (Abcam, ab4729), or anti-histone H3 antibodies (Abcam, ab1791). Chromatin DNA was purified by Dynabeads protein G (Invitrogen, 10004D) and subjected to real-time PCR. The region-specific primers are listed in Supplementary Table 1.

**DNA pull down assay.** To verify the results of ChIP assay, the DNA pull down assay was performed basically according to the manufacturer's protocol[51] with some modifications. Briefly, HSV-1-infected HeLa cells were fixed with 1% formaldehyde and sonicated on ice for 10 cycles at 30% power for 10 s with 20 s pauses between each cycle using a probe sonicator (Sonics & Materials, Inc.). Then, cell lysates were incubated with biotin-labeled HSV-1 ICP0 TSSs or scrambled sequence of ICP0 TSS4 and after washing, bound proteins were detected by western blotting with anti-H3K4Me3 antibodies (Abcam, ab8580), anti-H3K27Me3 antibodies (Abcam, ab6002), anti-H3K27Ac antibodies (Abcam, ab4729), or anti-histone H3 antibodies (Abcam, ab1791). Protein ratios of the indicated proteins/histone H3 were analyzed with ImageJ software.

**RNA immunoprecipitation assay.** The RNA immunoprecipitation assay was performed basically using the protocol of Peritz et al.[52] with some modifications. Briefly, HSV-1 or Mock infected HeLa cells were fixed with 1% formaldehyde and sonicated on ice for 2 cycles at 50% power for 15 s with 2 min pauses between each cycle using a probe sonicator (Sonics & Materials, Inc.). Then, cell lysates were incubated with anti-SRSF2 antibodies (Abcam, ab11826), anti-P300 antibodies (Abcam, ab59240), anti-CBP antibodies (Abcam, ab50702), anti-EZH2 antibodies (Abcam, ab186006), or anti-IgG antibodies (ab205718 or ab205719) at 4 °C overnight. RNA enriched from the immunoprecipitation was retrotranscribed for the real-time PCR. The primers for RT-PCR analysis are listed in Supplementary Table 1.

**Immunoprecipitation assay.** The immunoprecipitation assay was performed using the Immunoprecipitation Protein G Dynabeads® kit (Invitrogen, 10007D) according to the manufacturer's protocol. Briefly, HSV-1 or Mock infected HeLa cell lysates were incubated with antibodies against SRSF2 (Abcam, ab11826) and after washing, bound proteins were detected by western blotting with antibodies against PSPC1 (Abcam, ab104238), P54nrb (Abcam, ab70335), P300 (Abcam, ab59240), CBP (Abcam, ab50702), SRSF2 (Santa Cruz Biotechnology, sc-10252), EZH2 (Abcam, ab186006), or beta-actin (Proteintech, 60008-1-Ig).

**Statistics and reproducibility.** Each experiment was repeated three times. The results are presented as the mean ± SD. *$p < 0.01$. Comparisons between two groups were evaluated with a two-sample t-test. For three or more groups, standard one-way analysis of variance (ANOVA) followed by Bonferroni's test for multiple comparisons was completed. A two-tailed probability value <0.05 was considered statistically significant.

**Reporting summary.** Further information on research design is available in the Nature Research Reporting Summary linked to this article.

## Data availability

All data generated or analyzed during this study are included in this published article and its supplementary information files.

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

## Acknowledgements

This work was supported by the National Natural Science Foundation of China (32000878), and the Natural Science Foundation of Shandong Province (ZR2020LZL008).

## Author contributions

Z.W. designed the experiments. K.L. and Z.W. conducted the experiments, analyzed the data, and wrote the manuscript. Z.W. supervised the study.

## Competing interests

The authors declare no competing interests.
