## [Transparent Peer Review File · Communications Biology]

Reviewers' comments:

Reviewer #1 (Remarks to the Author):

The work reviewed here is clear and has a well-funded reasoning. The novelty of this work resides in the characterization of the main cellular components and complexes associated to the viral gene expression. The enzyme EZH2 and the P300/CBP complex seems to participate and regulate histone modifications on viral genes which promotes its transcription. The images are clear, the controls are presented properly. There is more than one experimental approach to validate every hypothesis. The text is well written. The only thing that I think deserves a second look is the addition of S5 and S6 to the main text. I think this addition will allow to the reader to follow the whole history more fluent. I'm sure this work will be a great contribution to the understanding how viral infection operates inside the cells.

Reviewer #2 (Remarks to the Author):

Viral infections elicit many changes to the host cell, either as pro-viral strategies or anti-viral responses. This study addresses the impact of herpes simplex virus 1 (HSV-1) infection on two dynamic subnuclear structures, speckles and paraspeckles that are usually in close proximity to each other and coordinate pre-mRNA processing.

Here the authors report that the cellular speckle (SRSF1) and paraspeckle components (NEAT1, PSPC1 and P54nrb) are redistributed in HSV-1 infected HeLa and C33A cells. The evidence consists of combined immunofluorescence and RNA FISH imaging. Unfortunately, a lack of careful quantitation or analysis at different times or virus inoculums makes the true extent of the redistribution hard to evaluate. The authors have previously reported the interaction of the splicing factor SRSF2 (Wang et al. 2016 JBC) and NEAT1 (Wang et al. 2017 CMLSci) with viral promoters. Using siRNA-mediated depletion they now propose that recruitment of SRSF1 is mediated by the ncRNA NEAT1 and in turn this leads to recruitment of the histone H3K27 methyltransferase EZH2 (associated with repression) and acetyltransferases p300/CBP (associated with activation). The functional significance of these associations and whether this is a consequence of infection stress is not addressed. Although the findings are provocative, the details are not fleshed out in any depth and in places there is a disconnect between the single-cell measurements and the bulk population assays.

SPECIFIC RECOMMENDATIONS

The redistribution is very difficult to assess without quantitation. The intensity plots are not helpful. Often single nuclei are shown but it is essential to quantify the changes over many infected cells and determine statistical significance in order to understand the impact of viral infection on any preexisting overlap between speckles and paraspeckles.

Importantly, the initial characterization of the infections shown in Supplementary figure 1 indicates that not all cells are productively infected. This greatly complicates the other analyses which invariably lack an internal marker for HSV-1 infection. The various biochemical assays, blotting and so on will most likely represent a mixture of infected and uninfected cells.

The impact of NEAT1 depletion on luciferase reporter expression is interesting but difficult to interpret. Why use promoter fragments in a transfected reporter rather than transcription of the real genes in the natural context of the viral genome? What is the basis for the selectivity of NEAT1 depletion on only two of the four promoters tested? The DNA pulldown experiment seems remarkably efficient but lacks essential specificity controls.

The biological relevance of H3K27me3 and EZH2 recruitment during productive HSV-1 infection is unclear especially at 4 hours post infection. The infections in Fig S6B difficult to interpret without the corresponding uninfected knockdowns.

In many places key experimental details are not explained and critical comparisons between uninfected and infected cells are not provided. This weakens many of the reported findings.

MINOR COMMENTS

The DAPI staining is inconsistent. In several figures the blue on the merge-DAPI panel is not visible at all. This would help identify locations of nucleoli and nuclear boundaries.

Why was 4 hours chosen as the infection end point?

The RIP experiment is not explained adequately. Is the RNA fragmented? Why does the antibody only recover one small portion of NEAT1v2?

Fig 1E lacks essential controls, such as other section of NEAT1v2 or unrelated DIG-labeled IVT RNA probes. Also was this analysis only performed on infected cells?

RIP and ChIP experiments are reported as 'fold enrichment of percentage input'. This obscures how well the assay works.

Details of infection conditions (amounts of virus, length of time, stain and source of virus etc.) are lacking in places. From the supplementary fig S1 it does not look like all cells are infected even though the stated MOI = 1.

In Fig S6A, the reduction in the histone modifications observed by immunofluorescence is not recapitulated by immunoblotting.

There are numerous typographical errors. For example, in Supplemental table 1 it should be 'sense' not 'sence'.

Reviewer #3 (Remarks to the Author):

Nuclear speckles are membraneless organelles that harbor splicing factors and play a role in RNA metabolism. The exact molecular function of paraspeckles has not been fully elucidated. The response to viral infection (innate immunity) is known to involve paraspeckles in the case of HSV-1.

Nevertheless, the mutual molecular interactions between nuclear speckles and paraspeckles in the response to infection have not been elucidated. This makes the findings by Li and al very exciting for molecular and cellular biology.

The study uses state-of-the-art imaging techniques and pulldown experiments, to show a novel interaction of nuclear speckle components with the paraspeckle main component NEAT1. This interaction is further demonstrated to be functional, as it changes the epigenetic environment of viral promoter regions in HeLa cells upon HSV-1 infection. These findings are novel because the implication of a cross-regulatory mechanism of nuclear speckles and paraspeckles in viral infection and the innate immune response. I have only a minor critique that most of the merged images do not show any DAPI staining, and some spelling error in "paraspeckles" (happens to me as well). I also recommend that the authors include an illustration with the envisioned model in the last figure.

Response to Reviewers

First of all, we thank both reviewers and editor for their positive and constructive comments and suggestions. Here are our responses to Reviewers as shown below.

Reviewers' comments:

Reviewer #1 (Remarks to the Author):

The work reviewed here is clear and has a well-funded reasoning. The novelty of this work resides in the characterization of the main cellular components and complexes associated to the viral gene expression. The enzyme EZH2 and the P300/CBP complex seems to participate and regulate histone modifications on viral genes which promotes its transcription. The images are clear, the controls are presented properly. There is more than one experimental approach to validate every hypothesis. The text is well written. The only thing that I think deserves a second look is the addition of S5 and S6 to the main text. I think this addition will allow to the reader to follow the whole history more fluent. I'm sure this work will be a great contribution to the understanding how viral infection operates inside the cells.

Response: According to reviewer's suggestions, we added Figure S5 and S6 to the main text (new Figure 3 and Figure 4).

Reviewer #2 (Remarks to the Author):

SPECIFIC RECOMMENDATIONS

1. The redistribution is very difficult to assess without quantitation. The intensity plots are not helpful. Often single nuclei are shown but it is essential to quantify the changes over many infected cells and determine statistical significance in order to understand the impact of viral infection on any preexisting overlap between speckles and paraspeckles.

Response: We followed the suggestions. To measure the degree of colocalization between channels, pearson's coefficient and overlap coefficient were quantified using the plug-in, JACoP in ImageJ software.

2. Importantly, the initial characterization of the infections shown in Supplementary figure 1 indicates that not all cells are productively infected. This greatly complicates the other analyses which invariably lack an internal marker for HSV-1 infection. The various biochemical assays, blotting and so on will most likely represent a mixture of infected and uninfected cells.

Response: We agreed with reviewer's concern. We checked carefully the Figure S1 and found almost each cell had the green signals representing HSV-1. In addition, we added negative controls using uninfected cells in critical experiments to confirm the effects of HSV-1 infection.

3. The impact of NEAT1 depletion on luciferase reporter expression is interesting but difficult to interpret. Why use promoter fragments in a transfected reporter rather than transcription of the real genes in the natural context of the viral genome? What is the basis for the selectivity of NEAT1

depletion on only two of the four promoters tested? The DNA pulldown experiment seems remarkably efficient but lacks essential specificity controls.

Response: In our previous study (J Biol Chem. 2016. 291(51):26377-26387; Cell Mol Life Sci. 2017. 74(6):1117-1131), we have found that knockdown of SRSF2 and NEAT1 reduced the levels of HSV-1 viral genes' mRNA and protein products using real-time PCR and western blotting, indicating that SRSF2 and NEAT1 regulate transcription of these genes. In this study, we constructed a luciferase reporter containing the promoter fragment of either ICP0 or TK gene to confirm whether SRSF2 and NEAT1 regulated the transcriptional activities of the ICP0 promoter and the TK promoter. ICP4 and ICP22 were used as negative controls. We added the information to the revised manuscript.

To determine the specificity in DNA pull-down experiment, we generated a scrambled sequence of ICP0 TSS4 and performed a DNA pull-down assay to examine the recovery of H3K4Me3, H3K27Me3, and H3K27Ac by scrambled ICP0 TSS4 fragment. The results showed that NEAT1 and SRSF2 have no significant effect on the recruitment of H3K4Me3, H3K27Me3, and H3K27Ac to scrambled ICP0 TSS-4 fragment (Figure S5).

4. The biological relevance of H3K27me3 and EZH2 recruitment during productive HSV-1 infection is unclear especially at 4 hours post infection. The infections in Fig S6B difficult to interpret without the corresponding uninfected knockdowns.

Response: In this study, we found that NEAT1 and SRSF2 could alter the histone modifications located nearby viral genes to modulate these genes' transcriptional activity. In these altered histone modifications, H3K4Me3 and H3K27Ac represent markers for actively transcribed genes, while H3K27Me3 is associated with gene repression. We think the role of NEAT1 and SRSF2 in viral genes' transcriptional activity is resulted from the comprehensive effect of these altered histone modifications. We added the information to Discussion section.

In addition, we added the uninfected knockdown controls for Fig S6B according to reviewer's suggestions (new Figure 4C).

5. In many places key experimental details are not explained and critical comparisons between uninfected and infected cells are not provided. This weakens many of the reported findings.

Response: We followed the suggestions. We added experimental details and critical controls using uninfected cells to the revised manuscript.

MINOR COMMENTS

1. The DAPI staining is inconsistent. In several figures the blue on the merge-DAPI panel is not visible at all. This would help identify locations of nucleoli and nuclear boundaries.

Response: We followed the suggestions. We provided DAPI staining in merge panel.

2. Why was 4 hours chosen as the infection end point?

Response: In the early hours of HSV-1 infection (0-8 hours post infection), HSV-1 genome will exploit host cell factors to facilitate its expression of immediate early gene and early gene (Arch Virol. 2012. 157, 1677 - 1688; Sci Rep. 2017. 7, 9176.). In addition, our previous study showed

that expression of NEAT1, the structural framework of paraspeckles, increased 2 hours post HSV-1 infection and peaked at 4 hours, gradually declining thereafter. Therefore, in this study, we chose 4 hours as the infection end point to investigate roles of speckles and paraspeckles in transcription of immediate early gene ICP0 and early gene TK.

3. The RIP experiment is not explained adequately. Is the RNA fragmented? Why does the antibody only recover one small portion of NEAT1v2?

Response: Yes, the RNA is fragmented. In RIP experiments, cells were fixed with formaldehyde to generate protein-RNA cross-links, and then sheared by sonication. Therefore, protein association with specific RNA regions can be assayed rather than the whole transcript. We illustrated it in Methods section.

4. Fig 1E lacks essential controls, such as other section of NEAT1v2 or unrelated DIG-labeled IVT RNA probes. Also was this analysis only performed on infected cells?

Response: We followed the suggestions. We prepared probes by transcribing fragments of NEAT1 N4 in vitro as a negative control and performed RNA-FISH assay in HSV-1 infected and uninfected cells.

5. RIP and ChIP experiments are reported as ‘fold enrichment of percentage input’ . This obscures how well the assay works.

Response: It means fold enrichment relative to input. To avoid misunderstanding, we change “Fold enrichment for % input” to “Fold enrichment (% of input)” in RIP and ChIP assays in the revised manuscript.

6. Details of infection conditions (amounts of virus, length of time, stain and source of virus etc.) are lacking in places. From the supplementary fig S1 it does not look like all cells are infected even though the stated MOI = 1.

Response: We followed the suggestions. We added the information to Figure legends and Methods section.

7. In Fig S6A, the reduction in the histone modifications observed by immunofluorescence is not recapitulated by immunoblotting.

Response: In original Fig S6A (new Figure S5), we generated a scrambled sequence of ICP0 TSS4 and performed a DNA pull down assay as a negative control for DNA pull-down assay in original Fig S5C and S5E (new Figure 3C and 3E).

8. There are numerous typographical errors. For example, in Supplemental table 1 it should be ‘sense’ not ‘sence’ .

Response: We are sorry for the mistake. We corrected this mistake in the revised manuscript. In addition, we have checked through our revised manuscript.

Reviewer #3 (Remarks to the Author):

Nuclear speckles are membraneless organelles that harbor splicing factors and play a role in RNA metabolism. The exact molecular function of paraspeckles has not been fully elucidated. The response to viral infection (innate immunity) is known to involve paraspeckles in the case of HSV-1. Nevertheless, the mutual molecular interactions between nuclear speckles and paraspeckles in the response to infection have not been elucidated. This makes the findings by Li and al very exciting for molecular and cellular biology.

The study uses state-of-the-art imaging techniques and pulldown experiments, to show a novel interaction of nuclear speckle components with the paraspeckle main component NEAT1. This interaction is further demonstrated to be functional, as it changes the epigenetic environment of viral promoter regions in HeLa cells upon HSV-1 infection. These findings are novel because the implication of a cross-regulatory mechanism of nuclear speckles and paraspeckles in viral infection and the innate immune response. I have only a minor critique that most of the merged images do not show any DAPI staining, and some spelling error in “paraspeckles” (happens to me as well). I also recommend that the authors include an illustration with the envisioned model in the last figure.

Response: We followed the suggestions. We added DAPI staining in merge panel and provided a schematic model of the role of speckle and paraspeckle in viral genes transcription in the revised manuscript (Figure 6E). In addition, We have checked through our revised manuscript to correct mistakes.